# Extraction and Physico–Chemical Characterization of Chitosan from Mantis Shrimp (*Oratosquilla nepa*) Shell and the Development of Bio-Composite Film with Agarose

**DOI:** 10.3390/polym14193983

**Published:** 2022-09-23

**Authors:** Suthasinee Yarnpakdee, Pimonpan Kaewprachu, Chalalai Jaisan, Theeraphol Senphan, Muralidharan Nagarajan, Sutee Wangtueai

**Affiliations:** 1Division of Marine Product Technology, Faculty of Agro-Industry, Chiang Mai University, Chiang Mai 50100, Thailand; 2Cluster of Innovative Food and Agro-Industry, Chiang Mai University, Chiang Mai 50100, Thailand; 3College of Maritime Studies and Management, Chiang Mai University, Samut Sakhon 74000, Thailand; 4Faculty of Engineering and Agro-Industry, Maejo University, Chiang Mai 50290, Thailand; 5Tamil Nadu Dr. J Jayalalithaa Fisheries University, Department of Fish Processing Technology, Dr. MGR Fisheries College and Research Institute, Ponneri 601 204, Tamil Nadu, India

**Keywords:** chitosan, mantis shrimp, deacetylation, agarose, composite film, barrier properties

## Abstract

Mantis shrimp (*Oratosquilla nepa*) exoskeleton, a leftover generated after processing, was used as a starting material for chitosan (CS) production. CS was extracted with different deacetylation times (2, 3 and 4 h), termed CS−2, CS−3 and CS−4, respectively, and their characteristics and antimicrobial and film properties with agarose (AG) were investigated. Prolonged deacetylation time increased the degree of deacetylation (DDA: 73.56 ± 0.09–75.56 ± 0.09%), while extraction yield (15.79 ± 0.19–14.13 ± 0.09%), intrinsic viscosity (η: 3.58 ± 0.09–2.97 ± 0.16 dL/g) and average molecular weight (M_ν_: 1.4 ± 0.05–1.12 ± 0.08 (×10^6^ Da)) decreased (*p* < 0.05). FTIR spectra of extracted CS were similar to that of commercial CS. Among all the CS samples prepared, CS−3 had the best yield, DDA, M_ν_ and antimicrobial activity. Therefore, it was chosen for the development of composite films with AG at different ratios (CS−3/AG; 100/0, 75/25, 50/50, 25/75 and 0/100). As the proportion of AG increased, the tensile strength (29.96 ± 1.80–89.70 ± 5.08 MPa) of the composite films increased, while thickness (0.056 ± 0.012–0.024 ± 0.001 mm), elongation at break (36.52 ± 1.12–25.32 ± 1.23%) and water vapor permeability (3.56 ± 0.10–1.55 ± 0.02 (×10^−7^ g m m^−2^ s^−1^ Pa^−1^)) decreased (*p* < 0.05). Moreover, lightness of the films increased and yellowness decreased. CS−3/AG (50/50) composite film exhibited high mechanical and barrier properties and excellent compatibility according to FTIR and SEM analyses. According to these finding, mantis shrimp exoskeleton could be used to produce CS. The developed bio-composite film based on an appropriate ratio (50/50) of CS−3 and AG has potential for being used as food packaging material.

## 1. Introduction

Mantis shrimp (*Oratosquilla nepa*), a marine crustacean species which is classified as fishery bycatch. They are abundant along the coast in the southern part of Thailand and have become economically important due to their palatability and useability in a variety of cooking. By the year 2021, the total of all species of mantis shrimp caught was 700–800 tons per year, with an estimated value of USD 1.18 million [1]. Processing of mantis shrimp generates large amount of leftover material, mainly exoskeleton. Generally, these byproducts have been used as fishmeal or fertilizer [2]. Increasing concern about pollution has stimulated an interest in converting byproduct materials to commercially valuable products. Around 40% of the mantis shrimps’ weight becomes waste, which is composed mainly of chitin (CT) or chitosan (CS), lipids, protein, calcium carbonate and pigment [2].

CT is a natural amino-polysaccharide mainly found in the exoskeletons of crustaceans, insects and the cell walls of fungi [3]. CT can be converted into its most well-known derivative, CS, through enzymatic or chemical deacetylation. CS is a linear polysaccharide composed of **β**-(1–4)-linked glucosamine units (2-amino-2 deoxy-D-glucopyranose) together with some proportion of N-acetylglucosamine (2-acetamindo-2 deoxy-D-glucopyranose) [4]. Currently, several studies are paying increasing attention to the refinement of CT and CS extraction and the development of materials. The properties of CS are affected by NaOH concentration, deacetylation time and temperature, and extraction technology [5,6]. Yaghobi et al. [7] reported that CS samples with different molecular weights (MW) and degrees of deacetylation (DDA) were prepared by controlling operating conditions throughout the multistage alkaline treatment. Sing et al. [8] stated that CT from squid pen was converted to CS with high DDA (86.55–89.72%) when the deacetylation temperature was raised to 130 °C. CS is one of the most abundant biopolymers after cellulose, which has been applied in many fields such as medicine, agriculture, food, textile, environment and bioengineering due to its excellent biological properties such as antimicrobial activity, nontoxicity, biocompatibility, biodegradability, chelating capability, etc. [9]. CS is a well-known biopolymer due to its film-forming ability, which has made important progress in food packaging.

Nowadays, biodegradable films are gaining increasing attention as important non-toxic and eco-friendly packaging materials versus synthetic thermoplastic films [9]. Most synthetic films are non-biodegradable and may cause environmental pollution and serious ecological problems. Different CS−based films have the potential to be employed as bioactive packaging due to their excellent gas barrier properties and fascinating antimicrobial properties, However, CS films have relatively poor mechanical properties and water vapor permeability (WVP) owing to the hydrophilic and macroporous nature of their CS structure. Blending other biopolymers with CS is expected to overcome these limitations. The blending of CS with other polysaccharides such as starch, cellulose, alginate and pectin have improved the properties of resulting films [10]. Wang et al. [11] found that CS blended with curdlan and carboxymethyl cellulose (CS/CD/CMC) exhibited better mechanical properties, permeability and thermal stability compared to pure CS. Agarose (AG), a biocompatible linear polysaccharide extracted from marine red algae, has the structure of 1,4-linked-3,6-anhydro-α-L- galactopyranose. AG has a distinct ability to form thermo-reversible gel, and its film is rigid through hydrogen bonding. It has been considered a favorable choice to manufacture composite films with great film-forming ability. Hu et al. [12] reported that addition of AG in varied proportions to CS solution enhances tensile strength (TS), elongation at break (EAB) and water vapor transmission rate (WVTR) of composite films. However, studies on CS extraction from mantis shrimp shells and its film-forming ability are scarce. Therefore, the objectives of this study were to investigate the effect of deacetylation times on extraction and to characterize CS from mantis shrimp shell, a byproduct of seafood processing. Composite films based on CS and AG at different proportions (100/0, 75/25, 50/50, 25/75 and 0/100) were also fabricated and characterized.

## 2. Materials and Methods

### 2.1. Chemicals/Bacterial Stains

All chemicals and microbial media used in the experiments were of analytical grade. Muller–Hinton agar was procured from Difco (Le Port de claix, France). Pathogenic bacteria *Bacillus cereus* (DMST 5040), *Staphylococcus aureus* (TISTR 2329), *Escherichia coli* (TISTR 527) and *Salmonella enterica subsp. Enterica serovar*. *typhimurium* (TISTR 2519) were obtained from the Department of Medical Sciences, Ministry of Public Health, Thailand (DMST) and Thailand Institute of Scientific and Technological Research (TISTR) (Bangkok, Thailand).

### 2.2. Collection and Preparation of Mantis Shrimp Shell

The semi-plain-dried mantis shrimp (*Oratosquilla nepa*) shell was gifted from a local entrepreneur in Satun Province, Thailand. Shells were packed in a polyethylene bag and transported to the Faculty of Agro-Industry, Chiang Mai University (Chiang Mai, Thailand) within 48 h. Upon arrival, the dried samples were washed using running tap water to remove the debris, followed by drying at 60 °C for 24 h in a tray-dry oven (Mextech, Seoul, South Korea). After drying, the samples were subjected to size reduction into a fine powder using a blender (Panasonic, Model MX-898N, Berkshire, UK). Crushed shells were then sieved to obtain a particle size of 1.0 mm and placed in an airtight polyethylene bag and stored in a refrigerator (4 °C) until use (within 1 month).

### 2.3. Preparation of CS 

CS was extracted from the shells of mantis shrimp as per the method by Knidri et al. [13] with a slight modification. Firstly, ground shell powder was demineralized using 3 M HCl (1:10, *w/v*) at 75 °C for 2 h with constant stirring using a magnetic stirrer. The samples were then filtered, washed with tap water for neutralization, and air-dried at room temperature for 30 min. The demineralized shells were deproteinized using 10% NaOH (1:10, *w/v*) at 80 °C for 2 h. The samples were then filtered through cheesecloth, washed with tap water until neutral pH was attained, and dried in an oven at 80 °C for 24 h. The obtained product was referred to as “chitin” (CT). The CT was then deacetylated with 50% NaOH with a ratio of 1: 20 (*w/v*) at 100 °C for various times (2, 3 and 4 h) to prepare chitosan (CS). The deacetylated CT was then filtered and washed with tap water until the pH of the washing water became neutral. The retentate was subjected to oven drying at 80 °C to obtain moisture content less than 10%, followed by grinding into a fine powder with particle size of 1 mm. The CS prepared by the different deacetylation times (2, 3 and 4 h) was referred to as CS−2, CS−3 and CS−4, respectively. The obtained products were placed in an airtight polyethylene bag and kept in a refrigerator (4 °C) until analysis, but not longer than 1 month. All CT and CS samples were weighed, and extraction yield and characterization were determined. The yield of CT and CS was determined by Equation (1):(1)$$\mathrm{Yield}\;\left(\%\right)=\frac{\mathrm{weight}\;\mathrm{of}\;\mathrm{CT}\;\mathrm{or}\;\mathrm{CS}\;\left(g\right)}{\mathrm{weight}\;\mathrm{of}\;\mathrm{starting}\;\mathrm{shell}\;\left(g\right)}\times 100\%$$

### 2.4. Characterization of CS

#### 2.4.1. Appearance and Color

A camera phone (iPhone 7 A1660, Apple Inc., Cupertino, CA, USA) was used to photograph the mantis shrimp shells, the CT and the CS.

Color of the CT and CS was determined using a colorimeter (ColourFlex, Hunter Lab, Reston, VA, USA) and reported in *L** (lightness), *a** (redness/greenness) and *b** (yellowness/blueness). Whiteness was calculated using Equation (2):(2)$$\mathrm{Whiteness}\;(\%)=100\%-\sqrt{{\left(100-L^{*}\right)}^{2}+{\left(a^{*}\right)}^{2}+{\left(b^{*}\right)}^{2}}$$

#### 2.4.2. Attenuated Total Reflectance-Fourier Transform Infrared (ATR-FTIR) Spectroscopic Analysis and Determination of Degree of Deacetylation (DDA)

CT and CS samples were analyzed by attenuated total reflection (ATR)-FTIR spectroscopy (Vertex 70, Bruker, Germany). Spectra were acquired in the IR range of 4000–400 cm^−^^1^. Automatic signals were collected for 32 scans at a resolution of 4 cm^−^^1^. DDA was determined using Equation (3) as described by Mittal et al. [14].
(3)$$\mathrm{DDA}\;(\%)=100\%-\left(\frac{A_{1320}}{A_{1420}}-0.3822\right)\times \frac{1}{0.0313}$$

#### 2.4.3. Determination of Intrinsic Viscosity [η] and Viscosity-Average Molecular Weight (M_v_)

Intrinsic viscosity [η] and viscosity-average molecular weight (M_v_) of CS were determined as described by Singh et al. [8]. CS was dissolved in solvent containing 0.2 M acetic acid and 0.15 M ammonium acetate to achieve final concentrations of 0.05, 0.1, 0.2, 0.3, 0.5 and 1.0 g/dL. Viscosity was measured at room temperature (25 ± 2 °C) using an Ostwald glass viscometer (Thomas Scientific, Swedesboro, NJ, USA). The time of flow for each CS solution from the upper mark to the lower mark was recorded. Reduction in viscosity [η_red_] was calculated using Equations (4)–(6):η_sp_ = η_r_ − 1(4)
(5)$$\eta_{r}=\frac{t}{t_{0}}$$
(6)$$\eta_{\mathrm{red}}=\frac{\eta_{\mathrm{sp}}}{C}$$
where η_sp_ = specific viscosity, η_r_ = relative viscosity, η_red_ = reduced viscosity, C = chitosan concentration (g/dL), t = flow time of CS solution, and t_0_ = flow time of the solvent. The η_red_ was plotted against CS concentration and [η] was calculated from a linear equation at zero concentration. 

M_v_ was calculated using the Mark–Houwink–Sakurada equation as Equation (7):[η] = KM_v_^α^(7)
where [η] = intrinsic viscosity, K = 9.66 × 10^−^^5^, and α = 0.742 as determined for the solvent at 25 °C [8].

#### 2.4.4. Determination of Antimicrobial Activity

The antimicrobial activity of CS was determined by using the broth dilution assay as performed by Wiegand et al. [15] and Lianga et al. [16].

The bacterial strains (*B. cereus*, *E. coli*, *S. aureus* and *S. typhimurium*) were grown in Mueller–Hinton soft medium broth (MHB) at 37 °C for 24 h. Suspensions were adjusted to achieve a turbidity equivalent of 0.5 McFarland (1–1.5 × 10^8^ colony-forming units (CFUs)).

The CS solutions (50 mg/mL in 1% acetic acid) were serially diluted two-fold using sterilized MHB to obtain concentrations ranging from 1.56–50 mg/mL. One milliliter of the dilutant was transferred to each tube of standard microbial suspension adjusted to 0.5 McFarland scale (1 mL). The samples were later incubated at 37 °C for 24 h. Microbial growth was measured in terms of optical density (OD) at 600 nm using a UV-1601 spectrophotometer (Shimadzu, Kyoto, Japan). The solvent (1% *v/v* acetic acid) was also measured in the same manner and did not inhibit microbial growth at the tested concentrations (≤1%). For a blank, CS suspensions for each concentration were run in the same manner without inoculum, and the values were subtracted from the test readings to avoid interference from the CS. The percentage of growth inhibition was expressed as the absorbance of solution containing diluted CS (A_t_) compared to the CS−free growth control as indicated below (Equation (8)).
(8)$$\mathrm{Growth}\;\mathrm{inhibition}\;(\%)=\frac{A_{c}-A_{t}}{A_{c}}\times 100\%$$
where A_t_ and A_c_ are the absorbance of the test group with blank subtraction and the control group, respectively. 

CS−3 showed high yield, DDA, M_w_ and antimicrobial activity and was selected for further CS/AG composite film preparation.

### 2.5. Preparation and Characterization of CS/AG Composite Films at Different Ratios

#### 2.5.1. Preparation of CS−3/AG Composite Film 

CS−3/AG composite films were developed according to the method of Hu et al. [2] with slight changes. To prepare a film-forming solution (FFS), CS−3 (1.5% *w/v* in 1% *v/v* acetic acid) was mixed with AG (1.5% *w/v* in hot distilled water) at different ratios (0/100, 25/75, 50/50/75/25 and 100/0). Thereafter, glycerol (30% *w/w* based on solid content) was added into FFS as a plasticizer. FFS was degassed using a sonicating bath (Elmasonic S 30 H, Singen, Germany) at 60 °C for 10 min. FFS (5.76 ± 0.01 g) was then cast onto a rimmed silicone resin plate (6 × 6 cm) and air-blown for 12 h at ambient temperature, followed by drying in a ventilated oven (Binder GmbH, Tuttlingen, Germany) at 25 ± 0.5 °C and 50 ± 5% relative humidity (RH) for 24 h. The resulting dried films were manually peeled off and subjected to analysis. Prior to testing, film samples were conditioned for 48 h at 50 ± 5% relative humidity (RH) and 25 ± 0.5 °C.

#### 2.5.2. Film Analyses 

##### Film Thickness 

The thickness of film samples of each treatment was measured using a hand-held micrometer (Mitsutoyo Co., Kanagawa, Japan). Twelve random positions around each film sample were determined. The average value was recorded.

##### TS and EAB

TS and EAB of film samples were investigated following the method of Kaewprachu et al. [17] by using a Universal Testing Machine (Strograph E-S, Toyo Seiki Seisaku-Sho Ltd., Tokyo, Japan). Five film samples (2 × 5 cm^2^) were tested using an initial grip distance of 3 cm and crosshead speed of 30 mm/min. After the films were pulled until broken, the maximum load and the final extension at break were used to calculate TS and EAB, respectively.

##### Water Vapor Permeability (WVP)

WVP was measured using a modified ASTM E96-80 method (ASTM, 1989) as described by Kaewprachu et al. [17]. A circle permeation cup filled with dried silica gel (0% RH) was covered with the film, placed in an environment chamber at 25 °C and 75 ± 5% RH, and weighed every 1 h for 12 h. Experiments were carried out in triplicate, and the results were expressed using Equation (9): (9)$$\mathrm{WVP}\;(g\;m\;m^{-2}s^{-1}\;{\mathrm{Pa}}^{-1})=\frac{\mathrm{Wl}}{\mathrm{tA}\cdot P}$$
where w is the weight gain of the cup (g), l is the thickness of film (m), t is the time of gain (s), A is the exposed area of film (m^2^), and ΔP is the difference in vapor pressure across the film.

##### Color and Transparency

The color of the film was determined using a Hunter Lab miniscan colorimeter as described in Section 2.4.2.

The transparency of film samples (40 × 40 mm) was measured by light transmittance at 600 nm using a spectrophotometer according to the method of Kaewprachu et al. [17]. The transparency of film was calculated using Equation (10): (10)$$Transparency\;value=\frac{-\log T_{600}}{x}$$
where *T*_600_ is the transmittance at 600 nm and *x* is the film thickness (mm). 

#### 2.5.3. Characterization of Selected Films 

A CS−3/AG composite film with an appropriate ratio (50/50) was selected for further ATR-FTIR and SEM studies rather than the films made with CS alone or AG alone. Prior to testing, all films were kept in an evacuated desiccator over fresh silica gel for 3 weeks at room temperature (28–30 °C) to obtain maximize dehydration.

##### ATR-FTIR Spectroscopic Analysis

The selected CS−3/AG composite films with ratios of 0/100, 50/50 and 100/0 were tested using the ATR-FTIR spectrometer as mentioned before (Section 2.4.3). Normalization was performed for all spectra before interpretation.

##### Morphology Study

The microstructure of the upper surface and cryo-fractured cross-section of the selected film was visualized using a scanning electron microscope (SEM) (Quanta 400; FEI, Praha, Czech Republic) at an accelerating voltage of 20 kV, as described by Nuthong et al. [18]. Analyses were carried out at 3500× magnification for surface observation, while the cryo-fractured films were observed at 3500× and 10,000× magnification for cross sections. 

### 2.6. Statistical Analyses

All experiments were run in triplicates (*n* = 3) by using a completely randomized design (CRD). Data were subjected to analysis of variance (ANOVA). Duncan’s multiple range test was performed for mean comparison at a 5% significance level (*p* < 0.05). Statistical analysis was by the Statistical Package for Social Sciences (SPSS 17.0 for Windows, SPSS Inc., Chicago, IL, USA) software.

## 3. Results and discussion

### 3.1. Extraction Yield 

CS was prepared by the following three steps: demineralization, deproteination and deacetylation. The effect of different deacetylation times on yield is shown in Table 1. CT was extracted from the intact mantis shrimp shell with a yield of 20.48 ± 0.72% after demineralization and deproteination. The result was in accordance with Jaganathan et al. [19], who reported average yield of CT from *Squilla* spp. of 24.18%. In general, crustacean shells consist of 30–40% protein, 30–50% calcium carbonate and phosphate, and 20–30% CT [20]. CT is found as a constituent of a complex network with proteins onto which calcium carbonate is deposited to form the rigid shell of crustacean [21]. Rojsitthisak et al. [22] reported higher CT yield from shrimp waste (20.0–27.0%) when demineralization was carried out prior to deproteinization. Microwave-assisted extraction of CT from *Oratosquilla oratoria* waste yielded 15.6% when alkaline protease and malic acid were used for deproteinization and demineralization, respectively [23]. The difference in the yield mainly depends on the conditions of the chemical extraction processes, including concentration of chemicals, soaking time and sequence of the treatments [22]. Under strong alkaline (50% NaOH) and high temperature (100 °C) conditions employed for deacetylation, hydrolysis of acetamido groups of acetylglucosamine in CT occurred. This resulted in formation of CS, a glucosamine polymer. As the deacetylation time of CT increased (2–4 h), the resulting CS yield decreased (14.13 ± 0.09–15.79 ± 0.19%) (*p* < 0.05). It was postulated that CS molecules were depolymerized during extraction, and smaller CS molecules were plausibly lost during the neutralization by washing. A similar trend was observed for CS extracted from squid pen [8], Pacific white shrimp shell [14], pink shrimp shell [5] and razor clam [6]. Moreover, NaOH concentration, deacetylation temperature and deacetylation time also directly impact the yield of CS [5]. Thus, deacetylation conditions, especially time, are crucial to determine CS yield.

### 3.2. Characterization of CS as Affected by Deacetylation Times

#### 3.2.1. Appearance and Color 

Appearance and color are the main parameters that impact the product’s external quality and consumer preferences. Color attributes (*L**, *a** and *b**) were used to determine the differences in color (Table 1). Shell powder had a chalky appearance with the lowest *L** (77.95 ± 0.07) and whiteness value (71.04 ± 0.16). This was in accordance with its highest *b**-value of 18.02 ± 0.34. A component in the shell might be the main contributor of its color. Normally, shells are rich in mixed compounds such as proteins, calcium carbonate, chitin, lipids and pigments [2]. Astaxanthin, a red carotenoid, has been identified as the predominant pigment in crustaceans [24]. Okada et al. [25] stated that three major forms of astaxanthin, i.e., diester, monoester and free forms, were the major components in the *P. monodon* exoskeleton, accounting for 86–98% of total carotenoids. In live mantis shrimp, astaxanthin is bound to a protein and crustacyanin as astaxanthin–crustacyanin complexes, giving a blue–green shell color. However, this complex is unstable with heat treatment. A pronounced increase in *a** value (9.69 ± 0.42) was observed for CT in comparison to that of raw shell (5.25 ± 0.08) and CS powders (3.03 ± 0.30–5.63 ± 0.49). CT powder turned orange–pink as visually observed. A higher *a** value suggested that carotenoid pigment is associated with CT to some extent. Generally, CT molecules are associated with protein chains via amide formation, usually between free amine groups in CT and sidechains of carboxylic groups in protein moieties [22]. High temperature and prolonged treatment time could destroy the chemical bond between CT and proteins. The solubilization of proteins under high alkalinity renders protein removal from the shell. Denaturation of carotenoproteins was induced, which resulted in the appearance of a red color due free carotenoid. The color saturation of CT depended on the amount of astaxanthin deposited. *L** and *b** values of CS samples were higher than those of CT sample. CS samples were visually off-white. CS−4 had the highest *L** and whiteness. The fading of the orange-pink color in CS was more pronounced when the deacetylation time increased (*p* < 0.05). This indicated that lightness and whiteness values increased with concomitant decreases in *a** and *b** values. It was suggested that the use of stronger alkaline concentration and a longer reaction time for the deacetyl group of CT could effectively eliminate the pigment–CT complex. This was reconfirmation that carotenoid in the crustacean shell is strongly bound to CT. Thus, stronger conditions are required to prepare an attractive whiter-colored CS [26]. Rasweefali et al. [27] reported that CS from deep-sea mud shrimp with 3 h of deacetylation time yielded a comparatively higher value of whiteness than those prepared with 1.5 and 6 h. However, Yen et al. [28] noted that longer reaction time had an adverse effect on the color of crab-based CS. Therefore, the deacetylation time significantly influenced the color of the obtained CS from mantis shrimp shell.

#### 3.2.2. ATR-FTIR and DDA 

ATR-FTIR spectra of shell, CT and CS from the mantis shrimp as affected by the deacetylation time (2–4 h) as well as commercial CS are depicted in Figure 1. FTIR spectra of original shell samples revealed the symmetric stretching vibration of hydrogen-bonded OH and amine (NH_2_) bands at 3457 and 3282 cm^−1^, respectively, while a band at around 2902 cm^−1^ represented C-H groups stretching. Amide-I, representing CO stretching from the acetyl group, was observed at 1658 cm^−1^. Absorbance peaks detected at 1400 and 867 cm^−1^ represent stretching and bending vibrations of CO_3_^2−^ (CaCO_3_; calcite) [29]. The spectrum features a peak at 1025 cm^−1^ that relates to the presence of C-O-C glycosidic bond. The presence of OH and NH groups occurred at 3461 and 3272 cm^−1^, respectively, for CT. A spectrum at 2892 cm^−1^ represents stretching vibrations of C-H (CH_3_ and CH_2_). The prominent spectra peak at 1632 and 1660, 1538, and 1320 cm^−1^ with characteristic stretching vibrations of amide I (C=O), II (N-H) and III (C-N) bands, respectively, for the CT sample. These bands are associated with the typical features of α-CT [30]. FTIR spectra of all CS showed absorption patterns corresponding to the characteristic of commercial CS (CC). The broad peak was detected between 3650 and 3000 cm^−1^ for CS samples. Due to intermolecular and intramolecular water traces, O–H and N–H peaks likely overlap to form the broad band that appeared in the same area for both groups. The band at 1558 cm^−1^ shifted to a higher wavenumber (1575–1592 cm^−1^) with increased deacetylation time. Moreover, the intensities of the amide I band (1660–1664 cm^−1^) and the amide-III band (1320 cm^−1^) also decreased with increasing deacetylation times, which confirms the occurrence of deacetylation of chitin (N-acetyl group). Mittal et al. [14] noted that a shift of bands to higher wavenumbers when deacetylation time increased. This was in accordance with the removal of the acetyl group from C-2 of CS, probably due to the increased breakdown of the H-bonds of the amide group. Singh et al. [8] noted that the intensity of peaks around 1645 and 1320 cm^−1^ decreased, while the peak at 1420 and 1590 cm^−1^ increased when the deacetylation temperature and time increased. The absence of bands at 1540, 1477, 1428, 877 and 725 cm^− 1^ of CT and CS extracted from mantis shrimp indicates the removal of protein and CaCO_3_ [23]. Thus, the CS prepared using different deacetylation times rendered different functional groups, particularly acetyl and amino groups. 

The DDA values from CS of mantis shrimp prepared using various deacetylation times (2, 3 and 4 h) are shown in Table 1 and ranged from 73.56 ± 0.09%−75.56 ± 0.01. The absorption ratio of 1320 to 1420 cm^−1^ peaks was used to calculate the DDA [8,14]. Brugnerotto et al. [31] reported that the A_1320_/A_1420_ ratio provides the lowest experimental error and is sensitive to the chemical composition of CS. A characteristic band at 1320 cm^−1^ measures the amount of N-acetylation (characteristic –OH, NH_2_, –CO groups), and the 1420 cm^−1^ band is suitable for comparison between D-glucosamine and N-acetylglucosamine. It was noted that DDA increased significantly with increasing deacetylation time (2–4 h). High temperature for extended time resulted in the excess removal of acetyl groups from the molecular chain of CT and the substitution of amino groups (NH_2_). A similar result was observed when CT from Pacific white shrimp shell was deacetylated at 110–130 °C for 2 to 4 h with DDA of 71.93–79.14% [14]. Nevertheless, harsh conditions for deacetylation, such as high temperature and longer time, might depolymerize CS chains, thereby liberating low-MW chitooligomers, which was associated with the decrease in yield. Therefore, an appropriate deacetylation time was necessary to achieve satisfactory CS yields with high DDA.

#### 3.2.3. Intrinsic Viscosity and Viscosity-Average Molecular Weight

The intrinsic viscosity (η) and viscosity-average molecular weight (M_ν_) of CS determined using the Mark–Houwink–Sakurada equation are shown in Table 1. The intrinsic viscosity of the obtained CS decreased (3.58 ± 0.09–2.97 ± 0.16 dL/g) as deacetylation time increased (*p* < 0.05). This result coincides with the decrease in M_ν_ (1.44 ± 0.05–1.12 ± 0.08 (×10^6^ Da)). In general, M_ν_ showed a direct correlation with intrinsic viscosity. The result was plausibly associated with the breakdown of the glucosamine backbone under high alkaline concentration (50% NaOH) during the deacetylation process at 100 °C for extended reaction time. Consequently, CS with lower molecular weight and reduced viscosity was obtained. This was confirmed by the increase in DDA with the increased deacetylation time. Similarly, Yen et al. [28] also reported that crab CT deacetylated with 40% NaOH at 105 °C for 60, 90 and 120 min resulted in CS with M_ν_ of 526, 513 and 483 kDa, respectively. Singh et al. [8] also observed a reduction in M_ν_ and viscosity of CS from squid pen with increasing deacetylation times and temperatures. Rasweefali et al. [27] found that CS extracted from deep-sea mud shrimp exhibited the highest DDA and lowest viscosity when a higher reaction time (6 h) at 100 ± 2 °C was applied. Thus, deacetylation time significantly influenced the structure of CS, especially M_ν_ and viscosity. 

#### 3.2.4. Antimicrobial Activity

Antimicrobial effects of prepared CS (CS−2, CS−3 and CS−4) at different concentrations (1.56–50 mg/mL) toward Gram-positive (Gram+) (*B. cereus and S. aureus*) and Gram-negative (Gram-) (*E.coli and S. typhimurium*) food-pathogenic bacteria are presented in Table 2. The inhibitory effect of CS was estimated by measuring the turbidity. The results showed that the growth of both bacteria types (Gram+ and Gram-) was inhibited by all prepared CS, indicating their broad inhibition spectrum. Their activity varied depending on deacetylation time and concentration used. Generally, CS is well-known for antibacterial activity. This could be attributable to the polycationic structure contained in CS (NH_2_-group), which can bind to the negatively charged (OH-group) bacterial cell membrane, thus leading to membrane permeabilization and inhibition of mRNA and protein synthesis via penetration of CS into the nuclei of the microorganism, and subsequent cell death [32]. In addition, Divya et al. [33] reported that CS acts as a chelating agent and electively binds to trace metal elements, causing toxin production and inhibiting microbial growth. The minimum inhibitory concentration of CS against pathogenic bacteria ranged from 3.13–12.50 and 6.25–12.50 mg/mL for Gram+ and Gram-, respectively. This shows that CS from mantis shrimp was more effective against Gram+ bacteria, especially *S. aureus*, than against Gram- bacteria. This was plausibly due to the differences on their cell membrane characteristics. Cell walls of Gram+ bacteria are a unique substance known as peptidoglycan, which is linked with teichoic acids. These are essential polyanionic polymers of the cell wall of Gram+ bacteria. The negative charge on the cell surface provided a molecular linkage with CS, leading to the disturbance of membrane functions [34]. Moreover, metal chelation by CS could affect cations on the membrane of Gram+ bacteria, thereby reducing the availability of metal ions essential for the survival of bacteria [35]. On the other hand, Gram- bacteria have only a thin layer of peptidoglycan and outer membrane with a lipopolysaccharide component, which possibly served as a penetration barrier against CS. The antibacterial effectiveness of CS on Gram- or Gram+ bacteria is unclear. In several studies, stronger antibacterial activity was apparent against Gram- bacteria than Gram+ bacteria [36,37]. Limam et al. [38] found that squilla CS had higher antibacterial effects against *E.coli* than *S. aureus*, while those obtained from pink shrimp CS showed the opposite effect. Among all prepared CS, CS−4 had the highest growth inhibition toward all pathogenic bacteria tested (*p* < 0.05) followed by CS−3 then CS−2, regardless of concentration used. The difference in antimicrobial activities could be attributed to the differences in intrinsic characteristics such as molecular weight and DDA [39]. The higher antibacterial activity of CS−4 was presumably due to the increase in the number of NH_2_ groups in CS with the higher DDA. The more interaction between CS and the cell membrane, the more the permeability of the cell membrane increases, resulting in cell leakage and further lysis. Moreover, the shorter chain of CS−4 might penetrate through the bacteria cell and bind to DNA easier than the longer one. Younes and Rinaudo [40] reported that the antimicrobial effect of high MW CS is caused by the formation of an impermeable layer around the cell, thus blocking the transport of essential solutes into the cell. Therefore, CS prepared from mantis shrimp with longer deacetylation times rendered higher DDA and low MW, playing a crucial role in the antimicrobial activity of CS. 

According to the obtained results, CS−3 had high yield, DDA, M_ν_ and antibacterial activity and was selected for further preparation of composite films by blending with AG.

### 3.3. Characterization of CS−3/AG Composite Films in Different Ratios

#### 3.3.1. Film Thickness

The thickness of blended films developed using different CS−3/AG ratios is given in Table 3. Film containing pure CS−3 (100/0) (0.056 ± 0.012 mm) had higher thickness than that containing pure AG (0/100) (0.024 ± 0.001 mm), while the thickness of CS−3/AG composite films was in the range of 0.037 ± 0.003–0.044 ± 0.003 mm (*p* < 0.05). No significant differences in thickness were observed between composite films prepared from CS−3 and AG (*p* > 0.05). However, thickness of composite films decreased as the AG ratio increased. CS−3 (DDA = 74.83 ± 0.34%) resulted in uneven distribution of D-glucosamine and N-acetyl-D-glucosamine residues that also possess some part of the acetyl group along with the polymeric chains. The complex group of CS−3 molecules aligned themselves in a looser fashion, thus providing a film structure with less compactness. The higher degree of compactness of linear polysaccharide might be because they align themselves to form an ordered network with less protrusion of the film matrix when AG was incorporated. This most likely resulted in the lower thickness of the obtained film. The result was in accordance with Cao et al. [41], who reported that the addition of CS into AG matrix significantly increased the thickness of blended films. However, El-Hefian et al. [42] found an increase in the thickness of CS film with increasing agar content. This might be due to the combined hydrophilic nature of the AG associated with water molecules, presumably involved in hydrogen bonding with those in CS. Therefore, the thickness of composite films was affected by CS−3/AG ratios.

#### 3.3.2. Mechanical Properties

The mechanical properties of composite films based on CS−3 and AG at different ratios are shown in Table 3. Tensile strength (TS) is the measurement of the maximum strength of a film against applied tensile stress, and elongation at break (EAB) represents the ability of a film to stretch [43]. Film made from CS−3 alone had lower TS (29.96 ± 1.80 MPa) but higher EAB (36.52 ± 1.12%) compared with CS−3/AG composite films (75/25, 50/50 and 25/75) (TS: 38.31 ± 1.13–54.06 ± 3.60 MPa and EAB: 25.27 ± 1.23–32.58 ± 1.81%) and AG films (TS: 89.70 ± 5.08 MPa and EAB: 25.32 ± 1.73%) (*p* < 0.05). TS of resulting films markedly increased with coincidental decrease in EAB when the ratio of AG increased (*p* < 0.05). Film made from CS−3/AG at a ratio of 50/50 had high TS (51.69 ± 1.25 MPa) and the highest EAB (32.58 ± 1.81%) among composite films (*p* < 0.05). The higher TS of composite films was more likely due to the formation of intermolecular H-bonds between the NH_2_− group of CS and the OH− group in the AG. The strength of hydrogen bonding increased and the arrangement of molecules was more ordered with increasing proportions of AG. This resulted in more compact and stronger films. However, the lower EAB of composite films indicated lower film extensibility. This might be due to the limited chain movement caused by the interactions of CS−3 and AG. The strengthening effect mediated by AG addition could reduce the extensibility of resulting films. Moreover, the decrease in EAB of composite films was plausibly due to the decrease in hydrophobicity of FFS as the mass ratio of CS decreased [41]. The decreases in weak bonds via hydrophobic interaction can stabilize the film matrix and decrease film extensibility. Thus, the AG ratio played a profound role in film network formation, which was governed by the chemical structure and impacted the mechanical properties of composite films. The addition of AG to CS−3 at a ratio of 50/50 could improve force required to break, with a negligible decrease in flexibility of the composite film. 

#### 3.3.3. Water Vapor Permeability (WVP)

The WVP of CS−3/AG composite films developed at different ratios is shown in Table 3. Pure CS−3 film had significantly higher WVP (3.56 ± 0.10 (×10^−7^ g m m^−2^ s^−1^ Pa^−1^)) than pure AG film (1.55 ± 0.02 (×10^−7^ g m m^−2^ s^−1^ Pa^−1^)), while CS−3/AG composite films had an intermediate value (2.40 ± 0.41–2.47 ± 0.08 (×10^−7^ g m m^−2^ s^−1^ Pa^−1^)). However, there was no significant difference in WVP among composite films (*p* > 0.05). In general, CS is amphiphilic in nature, containing both hydrophobic and hydrophilic regions. However, hydrophobic segments (alkyl chain) are shielded from water; the bulky hydrophobic region distributed within CS chains could enhance the free volume of the resulting film. As a result, more water and moisture could be absorbed easily into film protrusions. The stronger and more compact structure of AG film might effectively prevent the migration of water vapor. The result was in agreement with the increased thickness of the resulting films. Bourtoom and Chinnan [44] noted that the WVP of rice starch film increased with an increasing CS ratio. Fransiska et al. [45] also documented the augmentation of moisture content of CS films with decreasing concentrations of agar. In contrast, Hu et al. [12] found that the WVTR of composite films made from CS and AG increased as the ratio of AG increased. Actually, CS and AG are hydrophilic in nature due to their NH_3_^+^ and OH^−^ groups. Glycerol, a plasticizer, also contributes to hydrophilicity of the resulting film. Thus, CS and AG films had a lower moisture barrier property. 

#### 3.3.4. Color, Appearance and Transparency 

The color and appearance of the film are two main parameters for consumer acceptability and marketability. The lightness (*L**), redness/greenness (*a**) and yellowness/blueness (*b**) values of CS−3/AG composite films at different ratios are presented in Table 4. CS−3 films had the highest *b** value and the lowest *L** value compared with CS−3/AG composite films and the AG film (*p* < 0.05). Typically, commercial AG was transparent and colorless, while CS−3 had a slightly yellow appearance. The decreases in *b** and *a** values with coincidental increases in *L** value were noticeable when higher amounts of AG were incorporated. The changes in color of the resulting films were most likely attributed to the coloring components existing in CS−3. Pigment, such as the astaxanthin associated with mantis shrimp shell, plausibly leached out during extraction and might contribute to more yellowish color of CS−3 because the process was carried out without decolorization. In addition, the formation of yellow color in films was governed by the Maillard reaction [46]. The amino group (NH_2_-) in CS−3 could undergo a browning reaction along with carbonyl compounds (C=O) of reducing sugar in AG. Furthermore, lipid oxidation products associated with CS−3 more likely played a role in the yellow discoloration involved in the Maillard reaction. The result indicated that yellowness of composite film could be lowered by the incorporation of AG. This was possibly due to the NH_2_ precursors of film mixture being minimized by replacement with AG. Thus, the Maillard reaction could be retarded. Yang et al. [47] reported that the more yellowish color of cellulose–CS composite film was governed by an increased CS ratio. Garcia et al. [48] noted that corn starch–CS blend films changed from colorless to brown when the CS ratio increased. 

All films were visually flexible, homogeneous and smooth, and their surfaces had no visible pores or cracks. When the film sheets were placed on a white background, the color of the background was clearly observable. The lowest transparency value was obtained for CS−3 film (3.18 ± 0.05) compared to the others (3.29 ± 0.005–3.38 ± 0.002) (*p* < 0.05) (Table 4). The result suggested that CS−3 film was more opaque than the others. Higher transparency values were observed for CS−3/AG composite films when AG was added, which indicated the decreased transparency of the resulting films. Differences in the transparency of CS−3/AG composite films might be due to the differences in formation of browning reaction products. Kaewprachu et al. [17] also reported that films prepared from carboxymethyl cellulose (CMC) synthesized from young Palmyra palm fruit husk had low transparency when compared with the commercial CMC film due to their yellow color. Nagarajan et al. [49] also observed decreased light transmission and increased *b** value of films prepared from gelatin bleached with increasing concentration of H_2_O_2_. Therefore, blending CS−3 with AG could improve the color and transparency of the CS−3/AG composite films.

### 3.4. Characterization of the Selected CS−3/AG Composite Film

CS−3/AG composite film (50/50), which possessed better mechanical and water vapor barrier properties, appearance and color, was selected for further characterization in comparison to films made from CS−3 (100/0) or AG (0/100) alone.

#### 3.4.1. FTIR Spectra Analysis

FTIR spectra of CS−3, CS−3/AG at a ratio of 50/50, and AG films are illustrated in Figure 2. All films exhibited major peaks at 3230–3237 cm^−1^ (amide A, representative of N−H stretching coupled with hydrogen bonding) and 2861–2854 cm^−1^ (represents the stretching vibrations of C−H). The spectrum of CS−3 film is similar to previous reports [12,42], and the characteristic bands of CS−3 were clearly identified. The absorption peaks at 1612, 1542 and 1313 cm^−1^ correspond to amide I (representing C=O stretching of the acetyl group), amide II (representing N−H bending coupled with C−N stretching) and amide III (representing the vibrations in-plane of C−N and N−H groups of bound amide), respectively [14]. In addition, the absorption peaks at 1415 and 1369 cm^−1^ correspond to vibrations of OH and CH in the pyranose ring [50].

In the spectrum of AG film, the absorption peak at 1619 cm^−1^ was believed to be a feature of bound water presented in polysaccharides (O-H bending), while 1359 cm^−1^ represents CH_3_ bending of the alkane group [51]. The peaks around 1157 and 1139 cm^−1^ correspond to C–O–C bond stretching of the ether group of the agarose matrix. The characteristic absorption peaks at 925 and 885 cm^−1^ were ascribed to the 3,6 anhydrogalactose and C-H bending vibrations of anomeric carbon [12,43]. 

The changes in characteristic peaks of the composite film incorporating both components were noticed when CS−3 and AG were mixed. Amide II (N−H) and amide III (C−N) bands of the composite film (CS−3/AG: 50/50) were shifted to 1542 to 1544 cm^−1^ and 1313 to 1297 cm^−1^, respectively, compared to the CS−3 film. The decreases in these band intensities were obviously found when CS−3 replaced with AG. In addition, the peaks at 1162, 931 and 887 cm^−1^ were presented in CS−3/AG composite films. These results suggested that interaction between CS−3 and AG took place in the composite film matrix. The characteristic peak of inter- and intra-molecular H-bonds in CS−3 (3230 cm^−1^) and AG (3247 cm^−1^) were shifted to the intermediate frequency (3237 cm^−1^). The result was similar to findings of El-Hefian et al. [42], who reported that the shift of the OH and NH bands (3368 cm^−1^) of CS−based films to a higher vibrational wavenumber (3379 cm^−1^) was a result of the addition of agar up to 50%. The result reconfirmed that the presence of AG strengthened the H-bond interaction between molecules, resulting in compatibility of these two polymers in the composite films. This was in line with the higher TS, as shown in Table 3. Hu et al. [12] reported that differences in the composition of the films could be distinguished by the spectral shifts of the amide bands, and similar trends have also been reported by Cao et al. [41].

#### 3.4.2. Microstructure 

Figure 3 shows SEM micrographs of the surface (10,000×) and cryo−fractured cross-section (3500 and 10,000×) of selected CS−3/AG composite film (50/50) compared to CS−3 and AG film. AG film had a compact, smooth and homogenous surface, indicating an ordered film matrix. The roughness of the surface structure was more pronounced in CS−3 film than that found in AG film. The composite film possessed homogeneity, smoothness and no surface cracks, indicating complete miscibility between the two polymers. However, the composite film had a slightly coarser surface than the AG film. The differences in the microstructures of different films were caused by the varying arrangements of polysaccharide molecules during film formation. This was clearly observed for the fracture cross-section of CS−3 film, which had some discontinuous zones. This is in agreement with the poorer TS and higher WVP of CS−3 film compared to other films. The addition of AG to CS−3 contributed to a denser structure of the resulting film. The more compact structure of CS−3/AG composite films was mainly because of the reduced free volume within the CS−3 matrix, probably owing to intermolecular attractive forces and polysaccharide chain arrangement. The result was in accordance with the lower thickness of the composite film (Table 3). El-Hefian et al. [42] and Cao et al. [41] reported that the surfaces of blended films of CS and agar have no interface layer and are more homogeneous than those of pure CS and agar films. Thus, CS and AG blending can result in films with more compactness and density.

## 4. Conclusions

CS has been successfully extracted from mantis shrimp shell. The properties of the CS were different depending on the deacetylation time (2–4 h). With prolonged deacetylation time, more DDA occurred, with coincidental decreases in yield, [η] and M_ν_. The deacetylation time of 3 h rendered CS (CS−3) with good physico–chemical and antimicrobial properties, which encouraged film development. The properties of CS−3 film were improved by incorporating AG at a ratio of 50/50. Interactions between CS−3 and AG strengthened the film network and improved TS and barrier properties. However, it decreased EAB and yellowness of the resulting film. Thus, mantis shrimp shell could serve as a starting material for CS production. Bio-composite films based on CS−3 and AG with an appropriate ratio (50/50) could be used as food packaging.

## Figures and Tables

**Figure 1 polymers-14-03983-f001:**
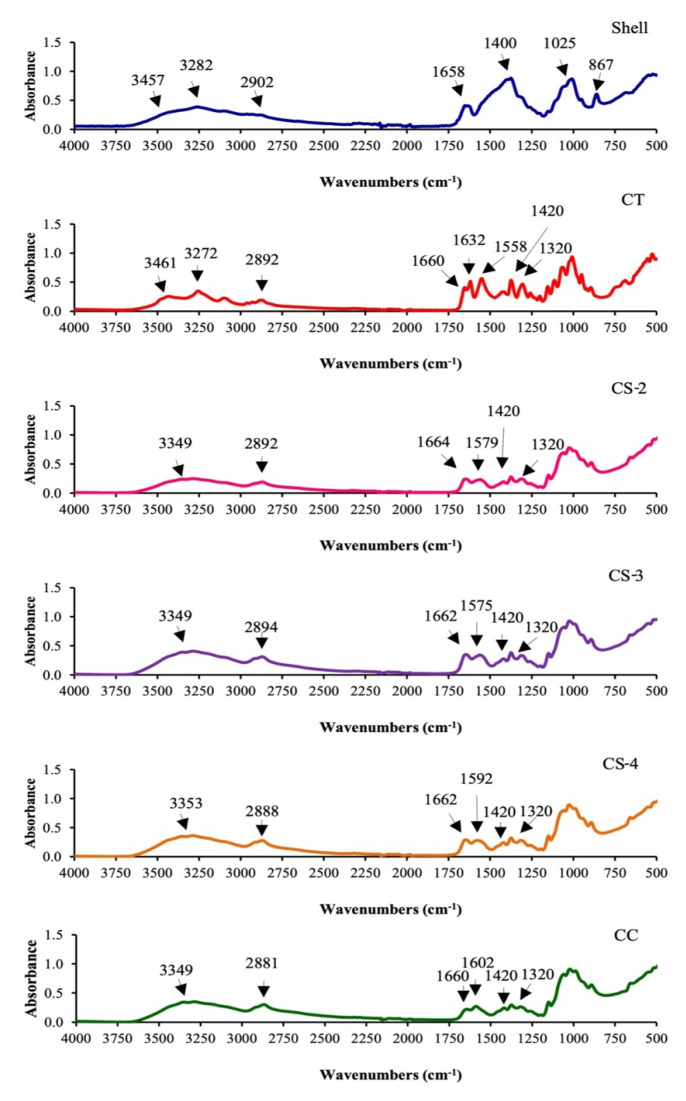
FTIR spectra of mantis shrimp (*Oratosquilla nepa*) shell, CT and CS with different deacetylation times (2−4 h). CT, chitin; CS−2, CS−3 and CS−4, chitosan prepared from chitin with deacetylation times of 2, 3 and 4 h, respectively; CC, commercial CS.

**Figure 2 polymers-14-03983-f002:**
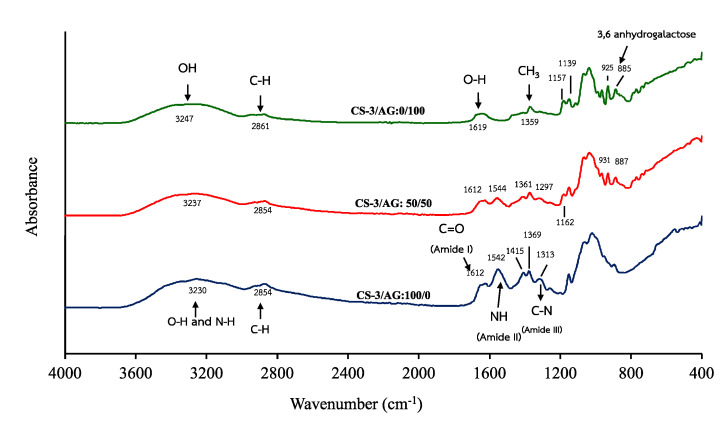
ATR−FTIR spectra of the selected CS−3/AG composite films with various ratios (100/0, 50/50 and 0/100).

**Figure 3 polymers-14-03983-f003:**
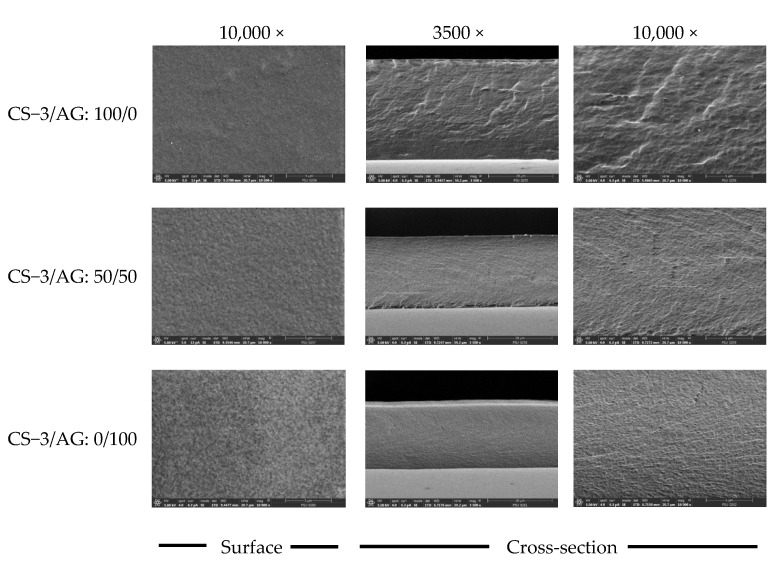
SEM micrograph surfaces and cross-sections of CS−3/AG composite films with various ratios. Magnification: 10,000× for surface; 3500× and 10,000× for cross section.

**Table 1 polymers-14-03983-t001:** Yield, color and physico–chemical properties of mantis shrimp (*Oratosquilla nepa*) shell, CT and CS obtained by different deacetylation times.

Parameters	Shell	CT	CS−2	CS−3	CS−4
Yield (%)	-	20.48 ± 0.72 ^a^	15.79 ± 0.19 ^b^	15.62 ± 0.25 ^b^	14.13 ± 0.09 ^c^
Appearance	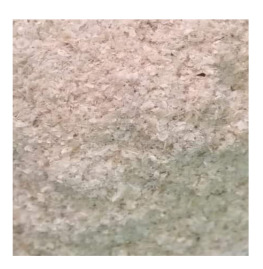	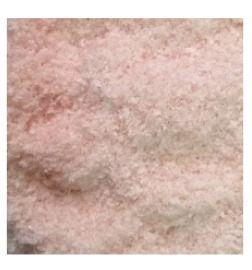	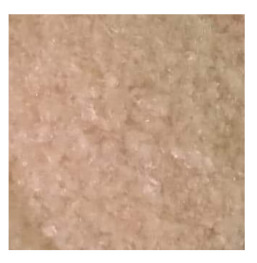	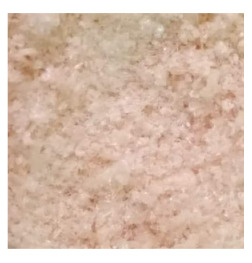	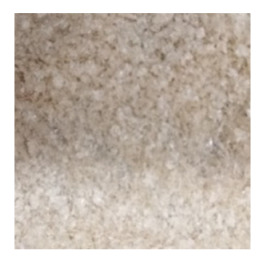
Color					
*L**	77.95 ± 0.07 ^d^	78.03 ± 0.40 ^d^	78.60 ± 0.24 ^c^	79.79 ± 0.48 ^b^	81.86 ± 0.09 ^a^
*a**	5.25 ± 0.08 ^b^	9.69 ± 0.42 ^a^	5.63 ± 0.49 ^b^	3.03 ± 0.30 ^c^	3.45 ± 0.20 ^c^
*b**	18.02 ± 0.34 ^a^	10.56 ± 0.20 ^d^	14.53 ± 0.47 ^b^	14.68 ± 0.68 ^b^	12.38 ± 0.42 ^c^
*Whiteness (%)*	71.04 ± 0.16 ^d^	73.76 ± 0.27 ^c^	73.53 ± 0.21 ^c^	74.83 ± 0.34 ^b^	77.76 ± 0.18 ^a^
Properties					
DDA (%)	-	-	73.56 ± 0.09 ^c^	74.83 ± 0.18 ^b^	75.56 ± 0.01 ^a^
[η] (dL/g)	-	-	3.58 ± 0.09 ^a^	3.44 ± 0.34 ^a^	2.97 ± 0.16 ^b^
M_ν_ (×10^6^ Da)	-	-	1.44 ± 0.05 ^a^	1.37 ± 0.18 ^a^	1.12 ± 0.08 ^b^

Values are mean ± SD (*n* = 3). Different lowercase superscripts within the same row indicate significant differences (*p* < 0.05). CT: chitin; CS−2, CS−3 and CS−4: chitosan obtained by deacetylation times of 2, 3, and 4 h, respectively.

**Table 2 polymers-14-03983-t002:** Antimicrobial activity of CS with different deacetylation times against different pathogenic bacteria by broth-dilution assay.

Type of Bacteria	Pathogenic Bacteria	Concentrations of CS (mg/mL)	Growth Inhibition (%)		
			CS−2	CS−3	CS−4
Gram+	*B. cereus*	50.00	24.90 ± 2.60 ^bB^	36.68 ± 3.51 ^aB^	37.95 ± 3.08 ^aB^
		25.00	16.41 ± 3.26 ^bB^	26.79 ± 4.93 ^aB^	29.27 ± 3.01 ^aB^
		12.50	7.94 ± 3.13 ^bB^	13.78 ± 3.68 ^abB^	17.02 ± 2.88 ^aB^
		6.25	1.66 ± 1.89 ^bA^	3.03 ± 2.22 ^bB^	8.50 ± 1.74 ^aB^
		3.13	-	-	6.71 ± 0.75 ^aB^
		1.56	-	-	-
	*S. aureus*	50.00	40.21 ± 4.17 ^bA^	52.81 ± 5.17 ^aA^	55.90 ± 2.58 ^aA^
		25.00	37.98 ± 3.37 ^bA^	46.22 ± 3.63 ^aA^	46.32 ± 2.10 ^aA^
		12.50	27.03 ± 4.03 ^bA^	28.70 ± 2.73 ^bA^	38.99 ± 1.70 ^aA^
		6.25	-	6.62 ± 1.34 ^bAB^	35.01 ± 0.87 ^aA^
		3.13	-	0.50 ± 0.38 ^bA^	13.20 ± 0.77 ^aA^
		1.56	-	-	-
Gram−	*E. coli*	50.00	31.42 ± 3.33 ^aB^	38.44 ± 5.59 ^aB^	38.54 ± 1.95 ^aB^
		25.00	13.14 ± 3.44 ^aB^	13.26 ± 5.97 ^aC^	16.85 ± 3.78 ^aC^
		12.50	11.25 ± 3.22 ^aB^	11.04 ± 4.41 ^aB^	15.25 ± 3.25 ^aB^
		6.25	-	9.55 ± 4.22 ^aA^	12.29 ± 5.65 ^aB^
		3.13	-	-	-
		1.56	-	-	-
	*S. typhimurium*	50.00	30.12 ± 7.04 ^aB^	30.66 ± 4.04 ^aB^	32.40 ± 5.24 ^aB^
		25.00	15.11 ± 6.00 ^aB^	17.93 ± 6.67 ^aBC^	21.34 ± 6.18 ^aC^
		12.50	8.81 ± 6.26 ^aB^	11.73 ± 4.14 ^aB^	17.66 ± 5.12 ^aB^
		6.25	-	7.27 ± 1.80 ^aAB^	8.58 ± 2.23 ^aB^
		3.13	-	-	-
		1.56	-	-	-

Values are mean ± SD (*n* = 3). Different lowercase superscripts in the same row within the same concentration level and pathogenic bacteria type indicate significant differences (*p* < 0.05). Different uppercase superscripts in the same column within the same concentration level indicate significant differences (*p* < 0.05). CS−2, CS−3 and CS−4: chitosan obtained by deacetylation times of 2, 3 and 4 h, respectively.

**Table 3 polymers-14-03983-t003:** Thickness, tensile strength (TS), elongation at break (EAB) and water vapor permeability (WVP) of CS/AG composite films with various proportions.

Composite Films (CS−3/AG)	Thickness (mm)	TS (MPa)	EAB (%)	WVP (×10^−7^ g m m^−2^ s^−1^ Pa^−1^)
100/0	0.056 ± 0.012 ^c^	29.96 ± 1.80 ^d^	36.52 ± 1.12 ^a^	3.56 ± 0.10 ^a^
75/25	0.044 ± 0.003 ^b^	38.31 ± 1.13 ^c^	29.09 ± 3.99 ^c^	2.40 ± 0.41 ^b^
50/50	0.040 ± 0.003 ^b^	51.69 ± 1.25 ^b^	32.58 ± 1.81 ^b^	2.51 ± 0.06 ^b^
25/75	0.037 ± 0.003 ^b^	54.06 ± 3.60 ^b^	25.27 ± 1.23 ^d^	2.47 ± 0.08 ^b^
0/100	0.024 ± 0.001 ^a^	89.70 ± 5.08 ^a^	25.32 ± 1.73 ^d^	1.55 ± 0.02 ^c^

Values are mean ± SD (*n* = 3). Different lowercase superscripts in the same column indicate significant differences (*p* < 0.05).

**Table 4 polymers-14-03983-t004:** Color attributes, appearance and transparency values of CS−3/AG composite films with various ratios.

Composite Films (CS−3/AG)	Color Attributes			Appearance	Transparency
	L*	a*	b*		
100/0	87.74 ± 0.69 ^c^	−1.32 ± 0.02 ^b^	3.09 ± 0.20 ^a^	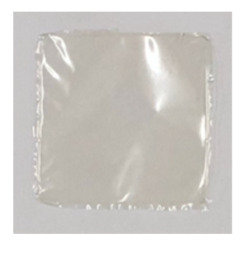	3.18 ± 0.005 ^e^
75/25	88.51 ± 0.04 ^b^	−1.39 ± 0.01 ^c^	2.28 ± 0.10 ^b^	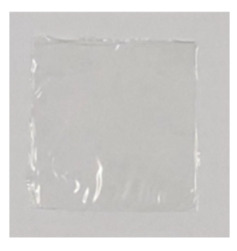	3.29 ± 0.005 ^d^
50/50	88.73 ± 0.07 ^b^	−1.42 ± 0.01 ^d^	1.61 ± 0.18 ^c^	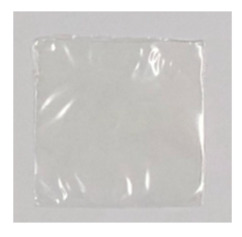	3.34 ± 0.002 ^c^
25/75	89.06 ± 0.03 ^a^	−1.44 ± 0.01 ^e^	1.02 ± 0.01 ^d^	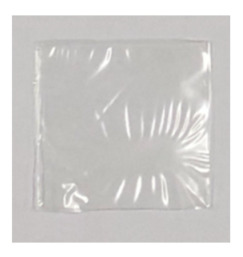	3.38 ± 0.002 ^b^
0/100	89.30 ± 0.21 ^a^	−1.2 ± 0.03 ^a^	−0.19 ± 0.03 ^e^	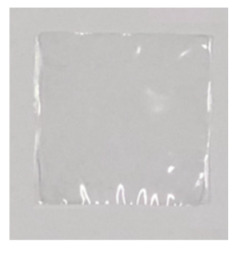	3.58 ± 0.002 ^a^

Values are mean ± SD (*n* = 3). Different lowercase superscripts in the same column indicate significant differences (*p* < 0.05). Numbers denote the ratio of CS−3 and AG.

## Data Availability

The data presented in this study are available on request from the corresponding author.

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
