# Peer review of "Extraction and Physico–Chemical Characterization of Chitosan from Mantis Shrimp (*Oratosquilla nepa*) Shell and the Development of Bio-Composite Film with Agarose"

_polymers, 2022, doi:10.3390/polym14193983_

Round 1

Reviewer 1 Report (New Reviewer)

I read very carefully the manuscript. I cannot suggest this paper for publication in a high-standard journal such as foods due to the following shortcomings:

- The major flaw is the lack of novelty, as the extraction of Chitosan under different conditions and the addition of agarose to CS have been well described and investigated in previous studies.

- The introduction is not focused and the problem is not well stated. 

- The scientific soundness of this paper is very weak. 

Author Response

Reviewer 2 Report (Previous Reviewer 3)

 The authors have made sufficient modifications according to the modification comments. Overall, the data of this paper is relatively substantial and the analysis is proper. The manuscript in its present version is apposite for publication in Polymers, and I suggest that this paper be accepted without further modification.

Author Response

Reviewer 3 Report (New Reviewer)

The Authors investigated the process of chitosan extraction from mantis shrimps, the influence of its parameters on the physicochemical properties of the product, and the possibility of using obtained chitosan in agarose films. The assumptions and results of relatively simple research have been presented in an understandable and orderly manner, supported by well-chosen literature references. The results of the research may have a high utility value related to the possible use of waste from seafood processing.

The authors should carefully review the article in terms of editing and remove errors related to, for example, incorrect spelling of words with uppercase or lowercase letters (e.g. “Bacterial”, “Typhimurium” in line 179, “Selected Films” in line 249, “windows” in line 272, “Selected” “Composite Film” in line 591, “Amide III” in line 615) and incorrect form of selected words (e.g. “become neutrality” in line 126, “bacterials” in line 422). 

The abbreviations TS and EAB (in line 222) used for the first time in the text should be preceded by the full name of the measured values "tensile strength" and "elongation at break". Authors should also not overuse these abbreviations in the text, as they are not commonly used terms for the mechanical properties of the materials tested. In turn, for the abbreviated spelling of the terms Gram-positive and Gram-negative bacteria, the terms "Gram+" and "Gram-" are commonly used.

The last remark concerns the possibility of improving the presentation of formulas (1), (2), (3), and (8). In these formulas "100%" should be given instead of "100" since the result is expressed as a percentage.

Author Response

Reviewer 4 Report (New Reviewer)

1.     Line no 45-47 doesn’t make sense, rewrite it.

2.     Reconsider the whole paper for grammatical mistakes and language errors.

3.     In line no. 73 and 74, it should be “gas barrier properties” and “fascinating antimicrobial properties”.

4.     “Therefore, the objectives of this study were to extract and characterize CS from 89 mantis shrimp shell as affected by deacetylation times.” Rewrite this line.

5.     Line no. 124 “The CT was then deacetylated with 123 50% NaOH with a ratio of 1: 20 (w/v) at 100 °C for various times (2, 3 and 4 h) to prepared 124 chitosan (CS).” It should be “prepare, not prepared”.

6.     Section 2.4.1 should be added in the experimental portion, rather than characterization.

7.     Again in the results and discussion, you have discussed extraction yield under the heading of characterization. Remove it from here.

8.     The headings should be written in a uniform format. Either in italics or straight.

9.     Improve the conclusion of your manuscript.

Round 2

Reviewer 1 Report (New Reviewer)

The manuscript has been improved after the revision. However, some minor points should be considered before acceptan:

Lines 60-61: Add the relevant references for "Currently, several studies paid more attention on the refinement of CT and CS extraction process and the development of material properties"

Lines 63-64: Define DDA and MW

Line 69: What is ‘antimicrobial nontoxicity’?!

Lines 80-81: Please revise the “The studies reported that blending of CS with other polysaccharides such as starch, cellulose, alginate, and pectin were potential for property improvement

Table 2:

-   It seems that the letters have been wrongly assigned to show statistical differences. So please revise.

-       Delete “CT: chitin;

Section 3.2.4: The discussion is contradictory regarding the antibacterial effectiveness of CH on Gram-positive and negative bacteria. Please revise.  

Author Response

This manuscript is a resubmission of an earlier submission. The following is a list of the peer review reports and author responses from that submission.

Round 1

Reviewer 1 Report

The paper is properly prepared. The authors have extracted chitosan from the shrimp exoskeleton and then used it in the form o chitosan and chitin for film purposes. The authors have made proper analyses of obtained films: mechanical properties, color, and transparency, water-solubility, and SEM micrographs. However, the topic of the research is not new. The problem of extraction of chitosan from shrimps was shown in: https://doi.org/10.3390/ma13215005 or https://link.springer.com/article/10.1007/s13201-019-0967-z

In my opinion, this work should be rejected because there is no novelty. 

Reviewer 2 Report

The work by Suthasinee Yarnpakdee et al. is concerned with the preparation and characterization and antibacterial properties of chitosan from mantis shrimps, as well as some properties of composite films of chitosan with agarose. Experimental work is done carefully, at a good level. However, I cannot recommend this paper for publication, as it is difficult to see any novelty comparing to the published research. Today, isolation and characterization of chitosan from various crustaceans is a routine work for hundreds of companies worldwide. The authors used the standard two-step chitosan production technology (demineralization + deacetylation) and obtained the expected characteristics of chitosan. The antibacterial properties of chitosan have also been studied in a huge number of papers. Although the topic of the manuscript is within the scope of Polymers, however, the paper looks more like a technical report. Thus, I recommend to reject this manuscript.

Reviewer 3 Report

In this paper, mantis shrimp (Oratosquilla nepa) exoskeleton, a left-over generated after processing, was used as starting material for chitosan (CS) production. Their characteristics, antimicrobial

and film properties with agarose (AG) were investigated. Overall, the paper has certain novelty and advantages for Research work in related field, and has value for publishing in Polymers. I suggest this manuscript can be published after the following minor revisions:

Line 242-243 Is it standard to test the transmittance at 600nm? Suggest additional supporting literature

Line 250-253 The humidity of the storage environment should have a non-negligible effect on the performance of the chitosan film, was the humidity controlled during storage? If yes, it is recommended to supplement the environmental humidity data.

Line 298-302 A large list of literature in the form of a review is not recommended. It is recommended to explain the possible reasons for the experimental results based on relevant literature and add the literature number at the end of the sentence.

Line 325 (9.69) suggest adding standard deviation, e.g. (9.69 ± 0.42), please test full text.

Line 367 and 369 “1,558” should be revised to “1558”; “1,320” should be revised to “1320”; “-1,558” should be revised to “1558”; “1,320” should be revised to 1320, please check the whole text and standardize the format.

Line 440 The “) mg/ml)" in Table2 needs to be corrected

Line 480- 482 (0.056), (0.024), “0.037-0.044” It is recommended to standardize the presentation format of numerical results, e.g. (0.05 6±SD).

Line 773, 782 and796 There is a lack of separation between journal name and year to match, please check the literature format and revise it according to the uniform form.
